# Metabolome–Microbiome Crosstalk and Human Disease

**DOI:** 10.3390/metabo10050181

**Published:** 2020-05-01

**Authors:** Kathleen A. Lee-Sarwar, Jessica Lasky-Su, Rachel S. Kelly, Augusto A. Litonjua, Scott T. Weiss

**Affiliations:** 1Channing Division of Network Medicine, Brigham and Women’s Hospital and Harvard Medical School, Boston, MA 02115, USA; rejas@channing.harvard.edu (J.L.-S.); rachel.kelly@channing.harvard.edu (R.S.K.); scott.weiss@channing.harvard.edu (S.T.W.); 2Division of Rheumatology, Immunology and Allergy, Brigham and Women’s Hospital and Harvard Medical School, Boston, MA 02115, USA; 3Division of Pediatric Pulmonary Medicine, Golisano Children’s Hospital at Strong, University of Rochester Medical Center, Rochester, NY 14612, USA; augusto_litonjua@urmc.rochester.edu

**Keywords:** microbiome, metabolomics, metagenomics

## Abstract

In this review, we discuss the growing literature demonstrating robust and pervasive associations between the microbiome and metabolome. We focus on the gut microbiome, which harbors the taxonomically most diverse and the largest collection of microorganisms in the human body. Methods for integrative analysis of these “omics” are under active investigation and we discuss the advances and challenges in the combined use of metabolomics and microbiome data. Findings from large consortia, including the Human Microbiome Project and Metagenomics of the Human Intestinal Tract (MetaHIT) and others demonstrate the impact of microbiome-metabolome interactions on human health. Mechanisms whereby the microbes residing in the human body interact with metabolites to impact disease risk are beginning to be elucidated, and discoveries in this area will likely be harnessed to develop preventive and treatment strategies for complex diseases.

## 1. Introduction

Trillions of microbes reside in the human body. The collective genomic contents (or microbiome) of the human population comprises upwards of 45 million non-redundant genes [1,2]. When compared to the human genome, which includes approximately 20 to 25 thousand genes [3], it is clear that the microbiota has the capacity to profoundly influence the biochemical environment of the host. Microbiomics refers to the study of all the genomes of the microorganisms present in an ecosystem, including eukaryotes, archaea, and bacteria [4]. The human microbiota performs essential biologic functions, including the harvesting of otherwise inaccessible nutrients [5], synthesis of vitamins [6], and development and maintenance of immune function [7]. One of the most famous discoveries of the Human Microbiome Project—which sequenced microbial genes at a variety of body sites in 300 subjects—was that while taxonomic composition varies between individuals, metabolic pathway abundances are relatively consistent [8]. Though this finding illustrates that “core” housekeeping pathways are stable across human microbiomes, variation in microbial metabolic potential does exist and may be important to the host response to environmental factors and disease states [8]. In fact, strikingly, because each person harbors rare microbial strains, approximately half of each person’s microbial gene content is unique, translating to a huge diversity in ways that microbes may produce or modify metabolites [1]. Indeed, microbial functions are closely reflected by the composition of the metabolome—that is, the collection of small molecules present in a sample. Although the human body houses many discrete microbiomes and metabolomes, the focus here will be on the gut, as this is taxonomically the most diverse and largest site [4].

In this review, we will give an overview of methods for integrative analysis of microbiome and metabolome data, discuss the growing body of evidence to support a strong relationship between the microbiome and metabolome, and how microbiome-metabolome interactions impact human disease.

## 2. Advances and Challenges in Integrative Analysis of the Microbiome and Metabolome

Advances in nucleic acid sequencing and informatics data analysis techniques now allow for comprehensive analysis of the microbiome [9,10]. Major techniques include marker gene and shotgun metagenomic sequencing. Marker genes, such as 16S ribosomal RNA for bacteria or Internal Transcribed Spacer (ITS) genes for fungi are present in microbes, but not in humans, and vary sufficiently across microbes to provide taxonomic information. Shotgun metagenomic sequencing involves sequencing all of the DNA present in a sample in short reads, which are then either assembled to approximate the original longer DNA sequence or matched to genes or species in a database. 

One technique to interrogate the function of the microbiota is to examine the genomic contents of the detected microbial taxa, which can either be directly measured by shotgun sequencing or inferred from the databases of known bacterial genomes. Identified genes can then be assigned functional and pathway annotations [11]. While this approach provides information about the functional potential of the microbiome, other techniques can directly evaluate the functional expression more. Several of these techniques were developed and are widely used for a broad range of research areas beyond the microbiome. For example, transcriptomics can be used to interrogate gene expression in a microbial environment in a metatranscriptomics analysis, and proteomics evaluates the structure, function, and interactions of proteins present in a sample including microbial-derived proteins [12]. A recent study highlighted the benefits of studying not only gene content, but protein abundances. Metagenomic sequencing and proteomic profiling of eight stool samples from a single subject with Crohn’s disease obtained over 4.5 years found that microbial gene abundances and protein abundances demonstrated divergent associations with laboratory measures relevant to inflammation in Crohn’s disease, and several biologically plausible associations were uniquely discovered by analyzing the metaproteome [13]. As we will discuss in this review, metabolomics, defined as the comprehensive analysis of small molecules in a sample measured by mass spectrometry or nuclear magnetic resonance, is especially valuable for yielding insights into the function of the microbiome and its impact on host health [14,15].

However, integration of human microbiome and metabolome data is challenging for several reasons [11]. In terms of choosing an appropriate analytic approach, researchers must grapple with the high dimensionality of these data in which metabolites and microbes typically vastly outnumber the sample size or number of individuals contributing data. Statistical methods must also account for the sparsity of microbiome data, in which many taxa are detectable in a minority of samples, as well as the compositional nature of both microbiome and untargeted metabolomics data, with abundances expressed relative to one another. Compositionality necessarily results from the unnormalized counts or intensities obtained from a DNA sequencer or mass spectrometer, respectively [16]. Compositional data is prone to bias and spurious associations can be detected if no steps are taken to correct this [16,17]. Approaches for addressing compositionality include adding control features with known concentrations to samples prior to profiling, data transformation such as centered log-transformation, and data normalization procedures, including recently developed techniques such as empirical Bayes normalization [16,17]. Interpretation of results can also be difficult because many detectable metabolites are of unknown chemical identity. Even among those with known identity, it is often unclear if a metabolite was ingested, produced by the host, a product of microbial metabolism, or resulted from some combination thereof. Databases of human microbial reference genomes serve as valuable resources [18,19] and the development of more comprehensive metabolite databases will be invaluable to future integrative analysis efforts. 

We will briefly discuss methods used for integrative analysis of microbiome and metabolome data. Further details have been recently reviewed elsewhere [20,21]. Multiple ordination methods utilize dimension reduction techniques to identify dominant features of one data set that co-vary with dominant features of a second data set obtained from the same samples or subjects. Canonical correlation [22], co-inertia analysis [23], and Procrustes analysis [24] are examples of multiple ordination methods. Network methods for integrating microbiome and metabolomic data include correlation analyses and can leverage pre-existing information by incorporating metabolic pathways to create edges between functionally related metabolites [20]. Associations with diseases and other outcomes can be sought using a variety of methods, including machine learning classifiers such as random forests [25] or support vector machines [26], as well as methods developed specifically for microbiome and other “omics”, such as Data Integration Analysis for Biomarker discovery using Latent cOmponents (DIABLO) [27]. This is an area of active investigation with new methods under development and recently published [28], including those utilizing neural networks [29]. For this reason, we are just beginning to understand the complex interrelationships between the human microbiome and metabolome and how they influence disease states. Future methods may identify new ways to leverage prior biological knowledge of metabolic networks and microbial phylogeny.

Of special interest are methods for ascertaining which metabolites are microbial-derived, and among those, which specific taxa are responsible for metabolite production. For example, a recently reported network method for determining the associations between metabolites and microbes in multi-omics data, when applied to metabolomic and metagenomic data from multiple studies, was able to assign metabolites to specific phylogenic clades that are presumably responsible for their production [30]. Similarly, the method Model-based Integration of Metabolite Observations and Species Abundances (MIMOSA) uses microbial gene sequencing data to estimate the community-wide metabolic potential for each metabolite and sample. Then, it compares this to metabolomics data from the same samples and estimates which species and genes are most important to the production of individual metabolites [31].

## 3. Interrelationships between the Microbiome and Metabolome and Their Impact on Human Health

The microbiome has a major impact on the metabolome, and as we discuss later in this review, this has relevance for human health. Mouse studies comparing germ free and colonized mice demonstrate the profound influence of the intestinal microbiome on the metabolome even at distant body sites including the kidney, liver [32,33], and plasma [34]. Comparing germ-free and conventional mice, hundreds of plasma metabolites in circulation are unique to conventional mice [34]. 

Mouse studies have shown that the intestinal metabolome is particularly tightly linked to the intestinal microbiome. Of 179 fecal metabolites, 70% significantly differed in abundance between germ-free mice and ex-germ-free mice who were inoculated with feces of specific pathogen-free mice [35]. Fecal and urine metabolomes also differ between germ-free mice, conventional mice, and mice with microbiota that were humanized using donor samples [36]. Introduction of even just one or two bacterial species to germ-free mice leads to shifts in the metabolome [36]. Similarly, antibiotic treatment perturbs the murine fecal and urine metabolomes [37]. 

Most recently, investigators comprehensively queried the impact of the microbiome on the metabolome throughout the body: the profiling of 750 samples from 96 sites in 29 organs from 4 germ-free and 4 colonized mice revealed that the microbiota affected the chemistry of every organ [38]. The gut metabolome was again found to be the most strongly influenced by the microbiome. Data from this study was used to identify previously unknown microbial amino acid amide conjugations to cholic acid [38]. The newly identified bile acid conjugates turn out to be present in humans and associated with cystic fibrosis and dysbiosis in Crohn’s disease, perhaps by way of activation of the Farnesoid X receptor (FXR).

These findings have been recapitulated in human studies, providing additional evidence that fecal metabolome composition is strongly influenced by the microbiome. In an analysis of colon samples from 47 human subjects, there was significant concordance between the cecal, and to a lesser extent, sigmoid metabolome and microbiome in terms of the variance and distribution of metabolites and microbes [39]. Correlation network analysis revealed widespread interrelationships between individual microbes and metabolites present in the gut [39]. Strong structural similarity between the gut metabolome and microbiome has been seen in other human populations, including the elderly [40] and in studies of Crohn’s disease [41] and pediatric asthma [42]. These findings were further borne out in a large study of 1116 fecal metabolites from 786 participants in the Twins UK study [43]. While host genetics only modestly influenced metabolite abundances, fecal microbial composition explained an average of 67.7% of observed variance in fecal metabolite levels. Tests of association between individual microbes and metabolites revealed rich connectivity between fecal microbes and metabolites in these twin subjects. 

The fecal microbiome has the strongest association with the fecal metabolome, and less robust but still significant associations with metabolomes distant from the gut, including urinary and plasma [36,42]. In an analysis of 659 plasma metabolites, 40 metabolites were identified which explained an average of 45% of the variance in fecal microbial Shannon index (an alpha diversity metric that reflects the number of species in a sample and how evenly abundances of those species are distributed) [44]. On the other hand, clinical laboratory tests, including cholesterol levels, electrolytes, liver and kidney function tests, white blood and red blood cell indices, circulating polyunsaturated fatty acids, and others were less predictive of fecal microbial alpha diversity [44]. Taken together, these findings reinforce the concept that resident gut microbes take part in determining the biochemical environment at sites near and far in the human body, and that the host metabolome in turn may influence gut microbial composition (Figure 1). The relationship between the microbiome and the metabolome is particularly intuitive given that these omics are influenced by shared upstream factors which include dietary, genetic, and environmental exposures [45,46,47,48].

The close relationship between the metabolome and microbiome has been leveraged in analyses of diseases that are thought to be caused, at least in part, by microbiome perturbations. For example, shifts in fecal microbiome and metabolome composition were observed in association with different phases of colorectal cancer [49]. In particular, the bile acid deoxycholate, which had been previously associated with tumorigenesis, was elevated in multiple polypoid adenomas and/or intramucosal carcinomas and co-varied with abundance of *Bilophila wadsworthia*, a species associated with inflammation. In a separate randomized trial of Mediterranean diet in 82 overweight and obese adults with low habitual intake of fruits and vegetables, the Mediterranean diet led to a decrease in fecal bile acids among other metabolite shifts [50]. Interestingly, those with the greatest reduction in bile acids had higher baseline fecal abundances of *Bilophila wadsworthia*, which also decreased with the Mediterranean diet intervention. These findings suggest that individualized features of the microbiome may impact the magnitude of benefit conferred by interventions such as adherence to a Mediterranean diet.

Profiling of the microbiome can also lead to personalized optimization of health-relevant metabolites. For example, glucose increases in response to meals were tracked longitudinally in 800 people and varied even when individuals consumed identical meals [51]. Fecal Proteobacteria and Enterobacteriaceae, which had been previously associated with components of the metabolic syndrome were positively associated with postprandial glucose after standardized meals. A predictive algorithm based on data including meal content (energy, nutrients), daily activity (meals, exercise, sleep), clinical laboratory values such as hemoglobin A1c and cholesterol, and microbiome features accurately predicted glucose response to real-life meals.

Future studies will benefit from more comprehensively querying the microbiome to including not only bacteria, but viruses, fungi, and archaea. For example, a study that challenged healthy but sedentary adults with a short-term exercise regime with or without daily whey protein consumption included profiling of fecal bacteria, viruses, and archaea. It found that whey protein consumption led to significant changes to fecal viral composition [52]. Interestingly, dietary-derived microbial metabolite trimethylamine N-oxide (TMAO), which has been associated with cardiovascular disease [53], demonstrated an intriguing and complex relationship with exercise and protein consumption.

## 4. Findings from Large Consortiums on the Metabolic Potential of the Gut Microbiome and its Impact on Health

The first phase of the Human Microbiome Project (HMP), which began in 2007, included 300 subjects who provided samples from a variety of body sites for microbial gene sequencing [8]. Microbial sequences were mapped to protein databases to infer the potential metabolic pathways represented in each sample’s collection of microbial genes. The findings of the HMP represented a major advance in our understanding of the diversity of organisms residing in the human body and how microbiome composition varies between body sites and between individuals. 

The more recent integrative Human Microbiome Project (iHMP), launched in 2012, has turned the focus towards (1) integrative omics including metabolomics, and (2) specific disease states that had been previously associated with the microbiome: prediabetes, inflammatory bowel disease, and preterm birth [54]. Resulting findings highlight the importance of microbe-metabolite interactions in health and disease. For example, the Integrated Personal Omics Profiling Study (iPOP), one of the iHMP studies, performed deep profiling of biological samples in 106 subjects, including stool and nasal microbiome profiling, peripheral blood monocyte transcriptomics, plasma metabolomics and proteinomics, and measurement of cytokines and growth factors in serum at multiple visits and one-time whole-exome sequencing [55]. Data from iPOP showed that several plasma metabolites were correlated with insulin resistance. Two networks were constructed based on correlations between individual fecal microbes and plasma metabolites: one for subjects with insulin resistance and one for subjects without insulin resistance. Correlations between microbes and metabolites differed between these two networks, suggesting that interactions between the host metabolome and gut microbiome are perturbed in insulin-resistant subjects. Differences were also observed by prediabetes status in multi-omic responses to physiologic stressors, including respiratory viral infection and influenza immunization. Interestingly, metabolomics outperformed other single “omics”, including transcriptomics, microbiomics, clinical data, and cytokine profiles in the classification of these stress states compared to health. While prediction based on all available “omics” combined performed best for respiratory infection, the use of all “omics” combined performed similarly to metabolomics alone for immunization.

In iHMP analyses of inflammatory bowel disease (IBD), fecal metabolites, and metabolic pathways were associated with IBD, including bile acids, short-chain fatty acids, sphingolipids, acylcarnitines, triacylglycerols, and tetrapyrroles [56,57]. A multi-omic network analysis of IBD-associated dysbiosis found that metabolites and metabolite classes demonstrated associations with microbial species and clinical laboratory parameters. Metabolome- and microbiome-based classifiers performed similarly in distinguishing subjects with and without IBD, a finding that was consistent with the strong association of these two “omics” in the gut [56]. A separate analysis found that differences between subjects with and without IBD were in fact most apparent in the fecal metabolome compared to the fecal metagenome, metatranscriptome, or proteome [57].

Metagenomics of the Human Intestinal Tract (MetaHIT, 2008–2012) was a European consortium that focused on the gut microbiome and its relation to obesity and IBD. Data from MetaHIT were used to identify metabolites, metabolite pathways, microbial genomic pathways, and bacterial microbes that associated with insulin resistance and metabolic syndrome. Specific species were identified that contributed to disease-relevant metabolic pathways, including *Prevotella copri*, abundance of which correlated with abundance of branched chain amino acids [58]. Taken together, findings from HMP and MetaHIT point to the importance of gut microbiota-metabolome interactions in disease states and highlight the relatively strong ability of the metabolome to predict disease states in comparison to other “omics”.

There are now well-characterized pathways in which members of the human gut microbiota influence metabolite abundances with relevance to human disease, even at distant body sites. For example, gut microbial metabolism of dietary choline results in increased plasma trimethylamine-N-oxide (TMAO), which has been reproducibly associated with development of cardiovascular disease [53]. Members of the gut microbiota may also modify responses to disease treatments: fecal microbiota composition influences the response to anti-PD1 cancer immunotherapy [59,60,61], and some medications such as metformin for type 2 diabetes mellitus lead to alterations in the gut microbiome that actually contribute to treatment efficacy [62]. The gut microbiome can also alter medication responses via drug metabolism, as in the case of L-dopa for Parkinson’s disease [63]. We recently reviewed microbial-derived metabolites and metabolite classes, including short-chain fatty acids, polyunsaturated fatty acids, and tryptophan metabolites that influence asthma pathophysiology and morbidity [64]. 

While it is clear that the gut microbiota influences a broad variety of human diseases through production or modification of metabolites, it remains an open question as to the relative importance of metabolite-mediated mechanisms compared to others, like the engagement of host receptors such as Toll-like receptors by pathogen-associated molecular patterns. Additionally, host-derived metabolites, cytokines, or other signaling molecules may impact microbiota composition as another mechanism of association between the microbiome and the metabolome.

## 5. Conclusions

Microbiome-metabolite interactions are pervasive in the human body. Recent evidence has revealed that the microbiome has a profound influence on the metabolome both near and at distant body sites, and the metabolome in turn can affect the microbiome. It is now clear that crosstalk between the microbiome and metabolome has an impact on many human diseases through a variety of mechanisms. This knowledge can be used to develop rational microbe- and metabolite-targeted treatments and preventive strategies, including probiotics, prebiotics and other dietary modifications, supplementing or inhibiting microbial-derived metabolites, and fecal microbiome transplant [65]. As understanding of metabolite–microbe interactions continues to improve, we expect additional insights to emerge to guide precision medicine approaches to health and disease. 

## Figures and Tables

**Figure 1 metabolites-10-00181-f001:**
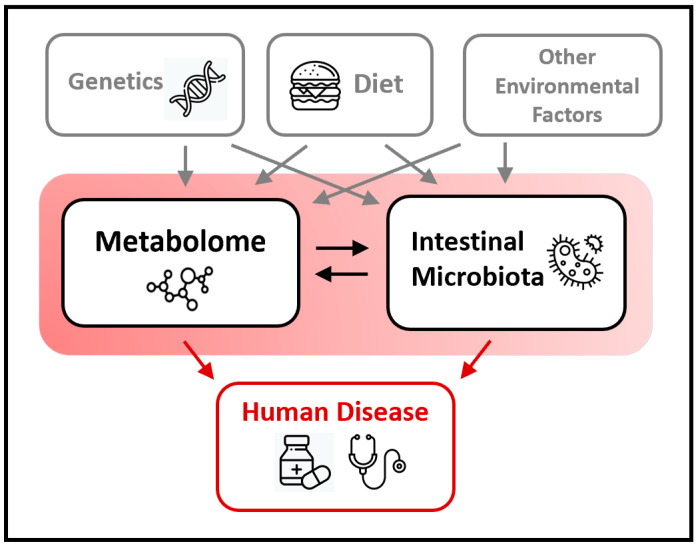
This simplified schematic illustrates two-way interactions between the metabolome and gut microbiome. Both the metabolome and microbiome are influenced by genetics, nutritional, and other environmental factors. Both also impact risk of human disease.

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
