# Peer review of "Metabolome–Microbiome Crosstalk and Human Disease"

_metabolites, 2020, doi:10.3390/metabo10050181_

Round 1

Reviewer 1 Report

The paper by Kathleen A. Lee-Sarwar et al. presents a review on the association between microbiome and metabolome. It focuses mainly on the human gut microbiome and the impact of microbiome-metabolome interactions on human health.

The topic involved in this review is of interest for the microbiome community.
Overall, the review is well written and well structured. It is quite short with only 37 references.

I found the review interesting. However, in my opinion it misses several relevant papers that need to be properly cited and discussed (probably in section 3). These are some examples:
- Yachida, S., Mizutani, S., Shiroma, H., Shiba, S., Nakajima, T., Sakamoto, T., Watanabe, H., Masuda, K., Nishimoto, Y., Kubo, M. and Hosoda, F., 2019. Metagenomic and metabolomic analyses reveal distinct stage-specific phenotypes of the gut microbiota in colorectal cancer. Nature medicine, 25(6), pp.968-976.
- Meslier, V., Laiola, M., Roager, H.M., De Filippis, F., Roume, H., Quinquis, B., Giacco, R., Mennella, I., Ferracane, R., Pons, N. and Pasolli, E., 2020. Mediterranean diet intervention in overweight and obese subjects lowers plasma cholesterol and causes changes in the gut microbiome and metabolome independently of energy intake. Gut.
- Cao, L., Shcherbin, E. and Mohimani, H., 2019. A Metabolome-and Metagenome-Wide Association Network Reveals Microbial Natural Products and Microbial Biotransformation Products from the Human Microbiota. MSystems, 4(4), pp.e00387-19.
- Cronin, O., Barton, W., Skuse, P., Penney, N.C., Garcia-Perez, I., Murphy, E.F., Woods, T., Nugent, H., Fanning, A., Melgar, S. and Falvey, E.C., 2018. A prospective metagenomic and metabolomic analysis of the impact of exercise and/or whey protein supplementation on the gut microbiome of sedentary adults. MSystems, 3(3), pp.e00044-18.

Reviewer 2 Report

The review work titled: " Metabolome-Microbiome Crosstalk and Human Disease" by Lee-Sarwar and colleagues discuss the increasing panel of works that link the microbiome with the metabolome elucidating new crosstalk interactions between them.

I find the paper well written, I only have some comments that should be addressed.

General comment

I find that the Introduction and the second chapter “Advances and Challenges in Integrative Analysis of the Microbiome and Metabolome” contains too few citations for a review work. I believe that in the first two chapters of a review work, and especially in the introduction chapter, most of the works under analysis should be pointed out.

Line 60-61: how shotgun sequencing can directly measures the functions of the microbiota?

I think this sentence needs to be clarified

Line 63: about the functional capacity

I think the word “capacity” should be revised with “potential”.

Line 64: functional reality. Several

I think the word “reality” should be revised with “expression”.

Line 94: and plasma[11].

A space should be added “plasma [11]”.

Line 107: from this study was

I think a citation here is missing.

Line 128: and how evenly abundances of those species are distributed)

I think this heavily depends on the alpha diversity measure used. I think either the sentence should be revised to be more general, or the alpha diversity measure should be specify.

Line 128-129: Clinical laboratory tests

I think here should be specified which clinical tests  the authors are referring to

Line 132: The connectivity between

I think the word “connectivity” should be revised with “relationship”.

Line 136: Figure 1. This simplified schematic illustrates strong two-way

I think the word “strong” should be remove.

Line 143: to infer the metabolic pathways

I think the sentence should be revised indicating that the metabolic pathways are from a functional potential and not functional expression.

Line 152: which deep profiling of biologic samples was

I think here should be specified (1) which “deep profiling” the authors are referring to, (2) which “biologic samples” the authors are referring to, and (3) double-check if “biologic” is the right wording, as I believe “biological” should be the correct wording here.

Line 160: including transcriptomics, microbiomics

I think microbiomics was never defined before as about what it should represent. I think this needs to be defined early on in the introduction.

Line 164-165: fecal metabolites and metabolic pathways were

I think here which metabolites and pathways should be defined.

Line 172: was a European

Should be “was an European”

Reviewer 3 Report

The authors present a succinct review on disease-associated multi-omic studies, as well methodologies, and some key findings that have been shown recently. In general, the text could benefit from discussing the work in-depth. Several points are presented and not analyzed at all.

General comments:

While it is beneficial to review the work of "large consortiums", there's other important multiomic work that has been published on "smaller scales", for example:

https://www.ncbi.nlm.nih.gov/pubmed/26590418

https://www.ncbi.nlm.nih.gov/pubmed/30801026

I suggest that the authors look into other published work like these two papers.

Line 77 would benefit from including references and discussion of work that tackles the issue of compositionality.

The discussion that starts in line 85 could benefit from concretely discussing and comparing the methodologies that are available. And also mention what are other potential directions where the field could move towards. This would help readers orient themselves on this rapidly evolving field.

Minor comments:

Line 124 should be revised.

Line 127 the alpha diversity metric determines the interpretation of the results. I suggest naming the metric used in the study.

Round 2

Reviewer 1 Report

The manuscript has been improved in this revised version and the authors have answered positively to my previous comments.